# MCMV Dissemination from Latently-Infected Allografts Following Transplantation into Pre-Tolerized Recipients

**DOI:** 10.3390/pathogens9080607

**Published:** 2020-07-26

**Authors:** Sahil Shah, Matthew DeBerge, Andre Iovane, Shixian Yan, Longhui Qiu, Jiao-Jing Wang, Yashpal S. Kanwar, Mary Hummel, Zheng J. Zhang, Michael M. Abecassis, Xunrong Luo, Edward B. Thorp

**Affiliations:** 1Department of Biomedical Engineering, Northwestern University, Evanston, IL 60208, USA; sahilshah2020@u.northwestern.edu; 2Department of Pathology, Northwestern University Feinberg School of Medicine, Chicago, IL 60611, USA; matthew.deberge@northwestern.edu (M.D.); y-kanwar@northwestern.edu (Y.S.K.); 3Comprehensive Transplant Center, Northwestern University Feinberg School of Medicine, Chicago, IL 60611, USA; andre.iovane@northwestern.edu (A.I.); sya016@northwestern.edu (S.Y.); qiulhui@mail2.sysu.edu.cn (L.Q.); jiao-jing-wang@northwestern.edu (J.-J.W.); m-hummel@northwestern.edu (M.H.); zjzhang@northwestern.edu (Z.J.Z.); 4Department of Surgery, Northwestern University Feinberg School of Medicine, Chicago, IL 60611, USA; 5College of Medicine, University of Arizona, Tuscon, AZ 85724, USA; mabecassis@medadmin.arizona.edu; 6Division of Nephrology, Department of Medicine, Duke University School of Medicine, Durham, NC 27705, USA; xunrong.luo@duke.edu; 7Department of Pediatrics, Northwestern University Feinberg School of Medicine, Chicago, IL 60611, USA

**Keywords:** transplant tolerance, donor specific transfusion, cytomegalovirus, latency

## Abstract

Transplantation tolerance is achieved when recipients are unresponsive to donor alloantigen yet mobilize against third-party antigens, including virus. After transplantation, cytomegalovirus (CMV) reactivation in latently-infected transplants reduces allograft viability. To determine if pre-tolerized recipients are resistant to viral dissemination in this setting, we transfused chemically-fixed donor splenocytes (1-ethyl-3- (3′-dimethyl-aminopropyl)-carbo-diimide (ECDI)-treated splenocytes (ECDIsp)) to induce donor antigen tolerance without immunosuppression. In parallel, we implanted donor islet cells to validate operational tolerance. These pre-tolerized recipients were implanted with murine CMV (MCMV) latently-infected donor kidneys (a validated model of CMV latency) to monitor graft inflammation and viral dissemination. Our results indicate that tolerance to donor islets was sustained in recipients after implantation of donor kidneys. In addition, kidney allografts implanted after ECDIsp and islet implantation exhibited low levels of fibrosis and tubulitis. In contrast, kidney cellular and innate immune infiltrates trended higher in the CMV group and exhibited increased markers of CD8^+^ T cell activation. Tolerance induction was unable to prevent increases in MCMV-specific CD8^+^ T cells or dissemination of viral IE-1 DNA. Our data suggest that latently-infected allografts are inherently more susceptible to inflammation that is associated with viral dissemination in pre-tolerized recipients. Thus, CMV latently-infected allografts require enhanced strategies to protect allograft integrity and viral spread.

## 1. Introduction

The current standard of care following transplantation is the use of broad spectrum immunosuppressants [1]. Despite tempering acute rejection with immunosuppression, late stage graft loss and susceptibility to infection remain a common problem. The administration of immunosuppression to promote engraftment is additionally associated with viral reactivation, hypertension, diabetes mellitus, anemia, and nephrotoxicity [2,3,4]. As an alternative to chronic immunosuppression, transplantation tolerance may be achieved in allograft recipients that are unresponsive to donor antigen but maintain reactivity to third-party antigens, including virus. Tolerogenic strategies include regulatory T cell infusion, donor-derived modified immune cells, and mixed chimerism via bone marrow transplantation [5,6,7]. Another tolerance approach is with chemically-modified donor specific transfusion (DST) [8,9,10,11]. Here, 1-ethyl-3-(3′-dimethyl-aminopropyl)-carbo-diimide (ECDI)-treated splenocytes (ECDIsp) render alloreactive T cells anergic via PD-1 signaling and induce regulatory T cell production [8,12]. Recently, ECDIsp-treated donor leukocytes with minimal short-term immunosuppression achieved islet graft tolerance in MHC class I-mismatched non-human primates [13]. Thus, tolerance-induction is an attractive goal not only to prevent graft rejection, but also for the maintenance of adaptive immunity to infection, such as from cytomegalovirus (CMV).

CMV is a member of the β-herpesvirus family that affects 30–97% of the population [14]. Its reactivation and dissemination is a major cause of morbidity after transplantation, particularly in cases of donor positive and recipient negative transplants [15,16]. With no successful vaccine, current therapy relies on the use of anti-viral prophylaxis and preventative care. However, there still remains the risk of late onset CMV, drug toxicity, the need for constant surveillance of viral DNA levels, and resistance of CMV to antivirals [16]. Recent studies in a clinically-relevant experimental model have shown that ischemia reperfusion injury from allograft implantation triggers reactivation of donor allograft CMV, and that continued administration of immunosuppressants is permissive for replication and dissemination of the virus [17]. This is particularly important for transplant recipients due to the administration of immunosuppression, which while promoting engraftment may also lead to an impaired anti-viral immune response. In a separate study of allograft tolerance-induction, pre-treatment with donor-derived cells was efficacious at inducing operational tolerance to CMV latently-infected donor allografts, as well as suppressing CMV dissemination [9]. However, the extent to which tolerance induction protects latently-infected allografts from inflammation and viral dissemination remains unclear.

In this study, we investigated the efficacy of the donor ECDIsp tolerance strategy in a full MHC-mismatch challenge of transplant tolerance. This was tested by implanting a second same-donor solid-organ allograft into an already tolerance-induced animal. Clinically-relevant latently-infected CMV donor kidneys were utilized as solid-organ allografts. Our data suggest that latently-infected allografts are inherently more susceptible to graft inflammation and that tolerance strategies should be optimized in cases where donor organs are latently-infected with CMV.

## 2. Materials and Methods

### 2.1. Experimental Animals

Eight- to ten-week-old *BALB/c and C57Bl/6* (B6) mice were obtained from Jackson Laboratory. All mice were housed under specific pathogen free conditions at Northwestern University’s Center for Comparative Medicine. All procedures used were performed according to guidelines from the Association and Accreditation of the Laboratory of Animal Care International (AAALAC) of the National Institutes of Health (NIH). Northwestern has an Animal Welfare Assurance with the Office of Laboratory Animal Welfare (A3283-01). All protocols (protocol number IS00001426) were approved by the Northwestern University Institutional Animal Care and Use Committee. 

### 2.2. CMV Infection and Establishment of Latency

Three week old *BALB/c* mice were infected intraperitoneally with 10^7^ plaque-forming units of the repaired Δm157 MCMV recombinant virus. The Δm157 MCMV virus was used to avoid MCMV resistance in C57BL/6 mice through natural killer (NK) cell Ly49H receptor and the viral m157 glycoprotein interaction [18]. The *CMV1^r^(Ly49h)* gene of NK cells confers resistance in C57BL/6 mice. The MCMV *m157* gene is the only MCMV ligand for the Ly49H receptor and the deletion of m157—in the case of the Δm157 virus—blocks Ly49H activation and results in higher virulence in C57BL/6 mice [19,20]. Additionally, our recombinant virus was repaired such that it did not possess a mutation in the *MCK2* gene, whereby this mutation can cause impaired viral trafficking [21]. Virus was purified as previously described, where BALB/c mouse embryonic fibroblasts from 15–17 day old embryos were cultured with salivary glands from acutely infected mice prior to collection and being tittered [22]. *BALB/c* mice were given 4–6 months to establish latency prior to use as kidney transplant donors, as previously described [17]. Latency was confirmed via serology. Virulence of Δm157 MCMV recombinant virus was measured following acute infection and compared to the MCMV Smith strain.

### 2.3. Islet Transplantation

B6 mice were treated with 170 mg/kg of streptozotocin (Sigma-Aldrich, St. Louis, MO, USA) to induce diabetes. Diabetes confirmation and islet transplantation methodology were as previously described [8]. Islets were transplanted by Longhui Qiu. Rejection of the graft was determined by consecutive blood glucose readings above 250 mg/dL.

### 2.4. Donor Splenocyte Transfusion

Islet tolerance was induced by intravenous injection of 10^8^ 1-ethyl-3-(3′ dimethylaminopropyl) carbodiimide (ECDI) coupled donor splenocytes (ECDIsp) at days −7 and +1, as previously described [8]. Briefly, 3.2 × 10^8^
*BALB/c* splenocytes were incubated with 30.75 mg ECDI in 1 mL of PBS on ice for 1 h with shaking followed by 3 washes and resuspension in PBS prior to injection. 

### 2.5. Kidney Transplantation

Donor negative or donor positive kidneys from *BALB/c* mice were transplanted into seronegative *B6* recipients with or without previous islet transplants as indicated. Transplant technical details have been previously described [23]. The recipient contralateral native kidney was not removed as this was the location of the original islet graft. Islet graft function was monitored via blood glucose measurements (OneTouch, Milpitas, CA, USA). Tissue samples were collected at a predetermined time point of 30–35 days and processed for flow cytometric, histologic, and viral analyses. Kidney transplant were performed by Jiao-Jing Wang. This time frame was chosen for collection as previous work using Δm157 MCMV latent donors resulted in high viral DNA levels in the allograft and ancillary organs at a similar time point [17]. 30–35 days allows enough elapsed time to monitor islet graft survival after kidney implantation and examination of the transplanted kidney.

### 2.6. Cell Isolation and Flow Analyses

Mice were euthanized and their kidneys were flushed with saline to remove cells remaining in the circulation. Tissue was digested with collagenase and DNAse at 37 °C for 45 min, and strained through a 40 μm filter. Erythrocytes were lysed and total cells quantified using a hemocytometer. Cells were incubated with Fc Block (Biolegend, San Diego, CA, USA) for 15 min and then stained for surface markers using fluorochrome-conjugated antibodies for 30 min on ice. For intracellular staining, Cytofix/Cytoperm (BD Biosciences, Rockville, MD, USA) buffers were used prior to staining with fluorochrome conjugated antibodies. Events were acquired using LSRFortessa (BD Biosciences, Rockville, MD, USA) and analyzed using FlowJo v10.4.2 (Tree Star LLC, Ashland, OR, USA). Antibodies and their specific clones used were: m38 tetramer (H2k^d^-m45, NIH Tetramer Core Facility), m45 tetramer (H2d^b^-m38, NIH Tetramer Core Facility). CD11b (M1/70), Ly6C (HK1.4), Ly6G (IA8), F4/80 (BM8), CD64(X54-5/7.1), Dectin-1 (BG1FPJ), DC-SIGN (MMD3), CD3 (17A2), CD4 (GK1.5), CD8 (KT15), CD8 (53-6.7), CD25(3C7), CD44 (IM7),PD-1 (29F.1A12), Foxp3 (FJK-16s), and Zombie Aqua Fixability Kit. 

### 2.7. Histology

Upon euthanasia and organ retrieval, islet and kidney graft sections were placed in 10% buffered formalin for 24 h, embedded in paraffin and sectioned into 4 μm slices. Kidney grafts were stained for (1) hematoxylin and eosin for leukocyte infiltration, (2) Periodic Acid-Schiff for tubulitis, (3) Masson’s Trichrome for fibrosis. Islet grafts were stained for insulin. Kidney sections were evaluated by renal pathologist (Y.S.K) for signs of acute and chronic rejection via leukocyte infiltration (interstitial and arteritis) and interstitial fibrosis. Severity of infiltration and fibrosis was scored from 0–4 with 4 being most severe [9]. Severity of tubulitis was scored by number of infiltrating monocytes in at least 2 tubules per field of view: 0 (no monocytes), 1 (<2 monocytes), 2 (2–4 monocytes), 3 (>4 monocytes, focal), 4 (>4 monocytes, diffuse) [23]. Additionally, PAS stained sections were quantified to determine the average number of infiltrating mononuclear cells per renal tubule per 200× magnified field of view for each mouse. For each mouse kidney section, 4 fields of view were assessed and at least 15 renal tubules were randomly chosen and analyzed per field of view to determine mononuclear cells infiltration. 

### 2.8. Viral DNA Analyses

Frozen tissue sections were processed to extract genomic DNA (Puregene, Qiagen, Valencia, CA, USA. Quantitative real time PCR (ABI Prism 7500, Foster City, CA, USA) was performed as previously described [24] to determine *MCMV* immediate early gene (*IE1)* copy numbers. Standard curves containing 10^2^,10^3^,10^4^, 10^5^, and 10^6^ copies of plasmid pIE111 and mouse kidney DNA were used to quantify *IE1* and *GAPDH*, respectively. Viral DNA copy number values were determined per one million cells for each samples by normalizing the MCMV IE-1 copies by the GAPDH copies for each sample and multiplying by 2 × 10^6^.

### 2.9. Statistical Analysis

All statistical analyses were performed using GraphPad Prism (Graphpad Software, La Jolla, CA, USA). A Shapiro-Wilk test was performed for all data sets to determine normality. For comparison of three or more groups either a non-parametric Kruskal–Wallis one-way ANOVA test with Dunn’s multiple comparisons or an ANOVA with Bonferroni’s multiple comparisons was utilized depending on data distribution. An unpaired *t*-test or Mann–Whitney Test was used to compare two groups. All data were displayed as a mean +/− SEM, with significance determined by a p-value less than 0.05.

## 3. Results

### 3.1. Experimental Approach and Induction of Operational Tolerance

The following series of experiments were outlined to determine the extent of allograft inflammation and murine cytomegalovirus (MCMV) allograft reactivation under conditions of recipient pre-established operational tolerance to alloantigen. To achieve this goal, our experimental approach sought to validate that transplant recipients were first indeed tolerant to alloantigen, prior to challenge with latently infected allografts. This was accomplished by two sequential transplantations, as follows.

For our strategy to induce antigen-specific tolerance, we chose a variation of Donor Specific Transfusion (DST), in which donor splenocytes are fixed with cross-linker ethylcarbodiimide (ECDIsp) [25] and pre-infused into selected recipients 7 days prior to implantation (day-7) and again at 1 day post implantation (day + 1). To first induce a state of operational tolerance, we administered *BALB/c* ECDIsp 7 days prior (-7) to implantation of *BALB/c* islet (I) cells (into B6 recipient kidney capsules) and 1 day post implantation (+1) into *B6 recipients*. Subsequently, MCMV latently-infected *BALB/c* kidneys were implanted into tolerance-induced *B6* recipients. To distinguish experimental groups, we utilized the following terminology: Latently-infected MCMV *BALB/c* kidneys were designated as “Kcmv” and non-latently-infected control kidneys as “K.” As a positive control for kidney allograft rejection, we utilized implanted Kcmv kidneys without ECDIsp treatment, designated Kcmv-Tx. Thus, we compared the following four groups: (1) non-transplanted latently-infected kidneys: Kcmv (2) latently-infected kidneys post-transplant: Kcmv-Tx (3) latently-infected kidneys post-transplant with initial pre-tolerance: Kcmv-Tx-ECDIsp and (4) non-latently infected kidneys post-transplant with pre-tolerance: K-Tx-ECDIsp. Only the ECDIsp-treated groups (Kcmv-Tx_ECDIsp and K-Tx-ECDIsp) received islet transplants. These experimental groups and the experimental structure are schematized in Figure 1 for reference. 

To first confirm function and operational tolerance of the initial islet implants, recipients were initially treated with streptozotocin (STZ) in order to destroy recipient insulin-producing β-cells [26]. Next, approximately 300 BALB/c islets were transplanted into B6 recipient mice (Appendix A). Following STZ -induced hyperglycemia and islet transplantation, *B6* recipients were confirmed to have corrected and sustained normoglycemia (data not shown) and as previously described [8,10,27], thereby establishing operational tolerance.

### 3.2. Are ECDIsp-Treated Islet Recipients Tolerant to Secondary Solid Organ Alloantigen Challenge with or without Latently Infected CMV?

To determine if *BALB/c* islet-implanted *B6* recipients were tolerant to subsequent challenge with a *BALB/c* solid organ allograft, we next transplanted MCMV seronegative recipients with a second allograft, consisting of a vascularized kidney from the same-donor strain *BALB/c*, at ~5 months post islet implant. We asked if the ECDIsp infusion strategy (day −7 and day +1, relative to islet implantation) was effective at maintaining primary allograft tolerance even after the stress of a subsequent kidney transplant. Figure 2A demonstrates that indeed STZ treatment triggered hyperglycemia prior to islet implantation. To measure islet graft tolerance, we performed immunohistochemistry of β-cell grafts that had been harvested ~30 days after transplantation of kidney allografts. Evidence of robust insulin staining was consistent with the maintenance of primary graft viability (Figure 2B). Functionally, recipient blood glucose measurements following kidney transplantation were normoglycemic, with no animals exhibiting hyperglycemic glucose levels above 250 mg/dL, independent of latent CMV infection (Figure 2C). These data indicate that early donor ECDIsp treatment was sufficient to protect islet grafts from the immunologic challenge of a subsequent solid organ allograft. 

We next directly evaluated the secondary kidney allografts after implanting in MCMV seronegative pre-tolerized recipients. We had chosen to examine the allografts at 30 days post renal transplant as a sufficient time frame to monitor viral replication, islet graft prolonged tolerance, and kidney graft survival. Because the primary islet grafts were initially implanted into a recipient kidney capsule (of the contralateral kidney to the subsequent kidney transplant), it was not possible to track functional parameters of renal function (such as blood urea nitrogen or creatinine) [17]. Instead, we focused our evaluation of kidneys by histology and flow cytometric analysis at the 30-day window post-kidney transplant. Compared to recipients that were not administered ECDIsp, ECDIsp-treated transplant recipients displayed significantly controlled graft weights (Appendix A). At the level of graft histology, interstitial fibrosis (as a marker of graft rejection) was decreased in the ECDIsp-treated groups compared to the non-ECDIsp-treated group (Figure 3A). A significant reduction in fibrotic mean score was measured between ECDIsp-treated non-latently infected (K-Tx-ECDIsp) and the kidney graft rejecting control (Kcmv-Tx). Compared to our positive control for graft rejection, Periodic acid-Schiff (PAS) and hematoxylin and eosin (H&E) staining consistently exhibited reductions in both tubulitis and mononuclear cellular infiltration in the ECDIsp-treated groups (Figure 3B,C). There were no significant differences in graft myeloid infiltrate between the ECDIsp-treated groups, but histological mean values were consistently lower for non-latently infected (K-Tx-ECDIsp) grafts. Closer examination of average numbers of mononuclear cell infiltration per renal tubule demonstrates a significant reduction of infiltrating cells in ECDIsp-treated groups compared to the rejecting control (Kcmv-Tx). Collectively, these data suggest that ECDIsp-treated islet recipients were also hypo-responsive to secondary challenge with donor alloantigen, and that latently infected allografts trended with a higher inflammatory signature.

### 3.3. Ecdisp-Treated Recipients Transplanted with Latently-Infected Allografts Exhibit Increased Trends of Innate Inflammation and Markers Of Macrophage (Mɸ) Activation 

By flow cytometry-assisted quantification of cells per milligram of kidney graft and percentage of total live cells, Ly6C^HI^ monocytes were found to be lower in the ECDIsp-treated non-latently infected recipients (K-Tx-ECDIsp) when compared to the non-ECDIsp-treated rejecting control (Kcmv-Tx) (Figure 4A). Graft F4/80-high macrophages were decreased in both ECDIsp-treated groups compared to the non-ECDIsp-treated rejecting control (Figure 4B). Once again, the CMV group trended with higher levels of monocytes and macrophages. In addition, we examined a mark of innate trained immunity, DECTIN-1, which has been linked to transplant rejection and inflammation [28]. Comparing ECDIsp-treated groups, latently infected grafts (Kcmv-Tx-ECDIsp) displayed an increased trend in DECTIN-1 expression on F4/80^+^ innate Mɸs (Figure 4B). Furthermore, DC-SIGN, which has been linked to viral infection, was significantly increased in latently infected (Kcmv-Tx-ECDIsp) graft Mɸs (Figure 4B) [29]. Collectively, the overall profile of innate immune cells and Mɸs in latently infected Kcmv-Tx-ECDIsp kidney allografts, relative to non-latently infected (K-Tx-ECDIsp) kidney allografts, was consistent with an elevated innate immune response. 

### 3.4. Ecdisp-Treated Recipients Transplanted with Latently-Infected Allografts Exhibit Reduced PD1+ CD8 T Effector Cells 

There were no significant differences in lymphocyte infiltrate between the ECDIsp-treated groups, but mean values were consistently lower for non-latently infected (K-Tx-ECDIsp) grafts (Figure 5A). ECDIsp treatment led to a significant decrease of CD4^+^ T cell frequency per mg of tissue and percent of total live cells relative to non-ECDIsp-treated recipient rejecting controls (Kcmv-Tx) (Figure 5A). Similarly, graft infiltrating CD8^+^ cell frequency of total live cells was significantly lower in the tolerance-induced group. We also examined the profile of specific T cell activation markers within the ECDIsp-treated groups. For example, higher frequencies of PD-1-expressing T cells have been shown to be critical for tolerance prolongation as a marker of T cell exhaustion [30]. In this context, latently infected (Kcmv-Tx-ECDIsp) grafts exhibited a decreased frequency of PD-1 expressing CD8^+^ cells (Figure 5B), consistent with a heightened T cell response. PD-1 expressing CD4^+^ cells in latently infected (Kcmv-Tx-ECDIsp) grafts also trended lower. 

### 3.5. Pre-Tolerized Recipients Were Permissive to Viral Dissemination

We next assessed markers of MCMV reactivation from latently-infected kidneys. As introduced above, we transplanted kidneys that were previously infected with Δm157 MCMV. Confirmation of latency in donors and prior to kidney transplant was assessed by measuring serum antibody production against MCMV (Supplementary Figure 3A), as previously detailed [17]. Acute infection determined comparable virulence in *C57BL/6* mice with Δm157 recombinant virus when compared to *BALB/c* mice infected with a well-established MCMV smith virus (Supplementary Figure 3B). In addition, as aforementioned, latently-infected MCMV *BALB/c* kidneys were designated as Kcmv and non-latently-infected kidneys as K. At the 30 day window post transplantation, we measured a significant increase in viral DNA copy numbers in the ECDIsp treated latently-infected (Kcmv-Tx-ECDIsp) grafts compared to contralateral, non-latently infected (K-Tx-ECDIsp) grafts, and non-ECDIsp-treated latently-infected rejecting control (Kcmv-Tx), consistent with viral reactivation and replication (Figure 6A). The non-transplanted latently infected kidneys (Kcmv) were the direct donor contralateral controls of the ECDIsp-treated latently-infected kidneys transplants (Kcmv-Tx-ECDIsp). No viral DNA levels were detected in the contralateral control (Kcmv), as ischemic injury is a major insult for viral gene transcription and replication, as noted by the high viral DNA copies of the Kcmv-Tx-ECDIsp group [31]. At the 30 day time point, the non-ECDIsp-treated rejecting control exhibited an intact anti-viral response as demonstrated by no detectable viral DNA levels following transplant. We detected viral DNA in the salivary glands and lungs of the ECDIsp treated latently-infected (Kcmv-Tx-ECDIsp) grafts, with particularly pronounced levels in the salivary glands (Figure 6B), indicating viral dissemination. The non-ECDIsp-treated rejecting control (Kcmv-Tx) did not present with any detectable viral levels in the lungs (Figure 6B). To identify specific immune markers of CMV reactivation, we utilized tetramer staining for recipient MCMV m38 and m45-specific CD8^+^ T cells. We observed an increase in both m38 and m45 MCMV-specific CD8^+^ T cells from the latently infected (Kcmv-Tx-ECDIsp) grafts (Figure 6C,D). Higher frequencies of persisting MCMV specific CD8^+^ T cells have been implicated as memory inflation and a marker of prior infection. The significant increase of m38 specific CD8^+^ T cells in the latently infected (Kcmv-Tx-ECDIsp) grafts compared to the contralateral non-transplanted donor (Kcmv) grafts suggests prior viral reactivation within the latently infected grafts. Collectively, these data indicate that pre-tolerance induction under our conditions was permissive to viral reactivation and inflammation within the graft, followed by viral replication and dissemination.

## 4. Discussion

Taken together, our findings indicate that MCMV latently-infected allografts are more susceptible to viral dissemination and exhibit heightened inflammatory trends following transplantation into pre-tolerized recipients. Early ECDIsp treatment was effective at protecting primary allo-islet cells, even after challenge with a second latently-infected donor organ. Although protection from viral dissemination was not conferred, early ECDIsp infusion remarkably induced a state of hypo-responsiveness to subsequent organ implantation, even though this occurred months after the initial ECDIsp infusion. We speculate that reactivation of latent CMV within the kidney graft is likely attributed to reperfusion injury during allograft implantation [17]. We propose that CMV latently-infected allografts require improved strategies to protect allograft integrity and prevent CMV dissemination.

At the level of the primary islet grafts, our data indicate stable intra-renal implantation, consistent with the lack of an allogeneic anti-islet response. ECDIsp-induced survival of two islet grafts in succession has previously been demonstrated [8]. Our data in this context independently corroborates the efficacy of ECDIsp during islet implantation and provides additional evidence of ECDIsp efficacy after challenge of additional donor vascularized grafts.

Within the second implantation group (kidney allografts), ECDIsp treatment proved superior to the non-tolerance-induced group, where untreated transplants exhibited increased evidence of tubulitis, and elevated levels of graft monocytes, macrophages, and CD4^+^ and CD8^+^ T cells. Despite this enhanced protection, both ECDIsp-treated cohorts exhibited heightened histologic scores of fibrosis, tubulitis, and inflammation, relative to the non-transplanted control. Elevations in inflammation could be explained by ischemia reperfusion injury following organ implantation [17], a potential low level alloimmune response, or viral cytokine expression [32,33,34]. Nevertheless, our data are consistent with a protective role of ECDIsp for graft prolongation through multiple transplants.

While ECDIsp was effective at inducing donor graft protection, recipients did not mount effective anti-CMV immunity, even though we measured increased levels of both m38 and acute phase m45 MCMV-specific T cells in the latently infected (Kcmv-Tx-ECDIsp) group, consistent with viral reactivation [35]. One possible explanation for this is reduced IFNγ production from MCMV-specific CD8^+^ T cells [36,37,38]. Our data may also be explained by ischemia reperfusion injury-based viral reactivation in the absence of immunosuppression [17]. Furthermore, separate studies utilizing ECDIsp infusion, but at the time of latently-infected kidney transplant, concluded that ECDIsp did not prevent viral reactivation, but instead blocked viral replication and dissemination [9]. A likely explanation for this finding is that in our experimental approach, ECDIsp was not administered at the time of implantation of latently-infected allografts. Importantly, we chose this pre-ECDIsp administration strategy because, in separate data not shown, we discovered that ECDIsp has the potential to directly modulate CMV replication. Therefore, we sought to uncouple direct effects of ECDIsp on the virus from its ability to induce allograft tolerance. 

At the cell and molecular level, our study revealed a unique signature that distinguished latently and non-latently-infected transplants after pre-tolerance induction. Latently infected Kcmv-Tx-ECDIsp kidney grafts had reduced PD-1-expressing CD8^+^ T cells, which is consistent with reactive effector T-cells and transplant rejection [39,40]. It will be interesting to determine in the future if decreased PD-1 cells are comprised of a population of exhausted anti-CMV CD8^+^ T cells that may facilitate viral reactivation. An alternative possibility to explain heightened graft inflammation is a decreased population of donor-specific tolerogenic cells. Within the myeloid compartment, we discovered a trend for increased macrophage expression of trained immunity marker Dectin-1. Nanobiologics that block inflammatory Dectin-1 have been shown to promote tolerance in a heart transplant model [28]. Separately, increased macrophage DC-SIGN in the latently infected (Kcmv-Tx-ECDIsp) group is consistent with another study in which renal allografts with elevated DC-SIGN^+^ dendritic cells were associated with acute transplant rejection [41]. The DC-SIGN receptor is implicated as a cell binding determinant for CMV and other herpes viruses [29,42,43]. 

Our study has important limitations to consider. This includes the potential for future experiments to provide enhanced statistical power to normalize data heterogeneity. This in and of itself is a significant challenge given the logistical complications of these complicated double transplant procedures, including due to recent and permanent disruptions from the COVID-19 coronavirus pandemic. An important limitation of our study—due to the extended time frame of surgeries—was that we did not directly isolate virus from the organs or measure viral RNA expression over multiple time points for a dynamic view of viral transcription and subsequent DNA replication. However, our high viral DNA copies in the ECDIsp-treated latently infected allografts (Kcmv-Tx-ECDIsp), while no virus was detectable in either the donor contralateral kidney (Kcmv) or latently infected rejecting control (Kcmv-Tx) affords reasonable confidence that the kidney allograft is the source of the viral genome. Additionally, the elevated MCMV specific CD8^+^ T cells in the graft suggest reactivation within the ECDIsp-treated latently infected grafts (Kcmv-Tx-ECDIsp). Another consideration relates to potential chronic endpoints after latent-CMV implantation. For example, our data suggest increased inflammation in the latently-infected group. Thus, it may be informative in future experiments to examine later stage pathophysiology and vasculopathy between non-latently infected (K-Tx-ECDIsp) and latently infected (Kcmv-Tx-ECDIsp) transplants. Future experiments may also examine the contribution of additional anti-donor myeloid and lymphoid cell subsets. For example, previous transplant models using ECDIsp have implicated expansion of T_regs_ and myeloid derived suppressor cells [8,44]. We also cannot rule out potential contributions of B-cells and donor reactive antibody responses to the graft.

In conclusion, we report MCMV dissemination after transplantation of latently infected kidneys into pre-tolerance-induced recipients. ECDIsp treatment remains an attractive approach to optimize donor specific tolerance, as for example it does not require ex vivo expansion of T_regs_ or the risk of graft versus host disease during mixed chimerism [5,45], although these later approaches still also hold significant promise. Future studies are warranted to optimize ECDIsp or DST therapy in the setting of CMV reactivation and to determine to what degree viral reactivation and dissemination directly impacts allograft tolerance.

## Figures and Tables

**Figure 1 pathogens-09-00607-f001:**
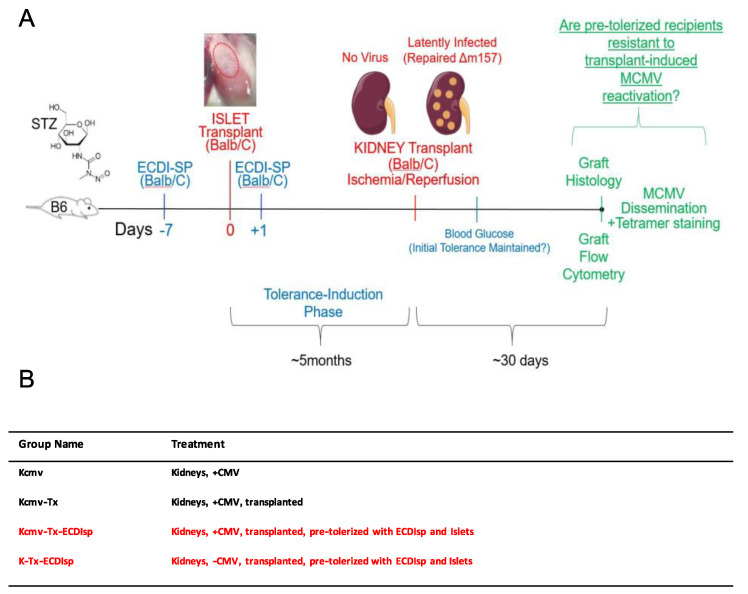
Experimental approach. (**A**) Donor splenocytes were fixed with cross-linker ethylcarbodiimide (1-ethyl-3- (3′-dimethyl-aminopropyl)-carbo-diimide (ECDI)-treated splenocytes (ECDIsp)) and pre-infused into selected recipients 7 days prior to islet implantation (day-7) and again at 1 day post implantation (day + 1). Streptozotocin (STZ) was administered in order to induce type 1 diabetes. We administered *BALB/c* ECDIsp 7 days prior (−7) to implantation of *BALB/c* islet cells (into B6 recipient kidney capsules) and 1 day post implantation (+1) into B6 recipients. For the viral arm of our study, latently-infected murine cytomegalovirus (MCMV) *BALB/c* kidneys were designated as “Kcmv” and non-latently-infected control kidneys as “K”. As a positive control for allograft rejection, we implanted Kcmv kidneys without pre-tolerance, designated Kcmv-Tx. Thus, we compared the following four groups: (1) non-transplanted latently-infected kidneys: Kcmv, (2) latently-infected kidneys post-transplant: Kcmv-Tx, (3) latently-infected kidneys post-transplant with initial islet allograft pre-tolerance: Kcmv-Tx-ECDIsp, and (4) non-latently infected kidneys post-transplant with initial islet allograft pre-tolerance: K-Tx-ECDIsp. (**B**) Shorthand abbreviation and description of the different treatment groups and control for this study.

**Figure 2 pathogens-09-00607-f002:**
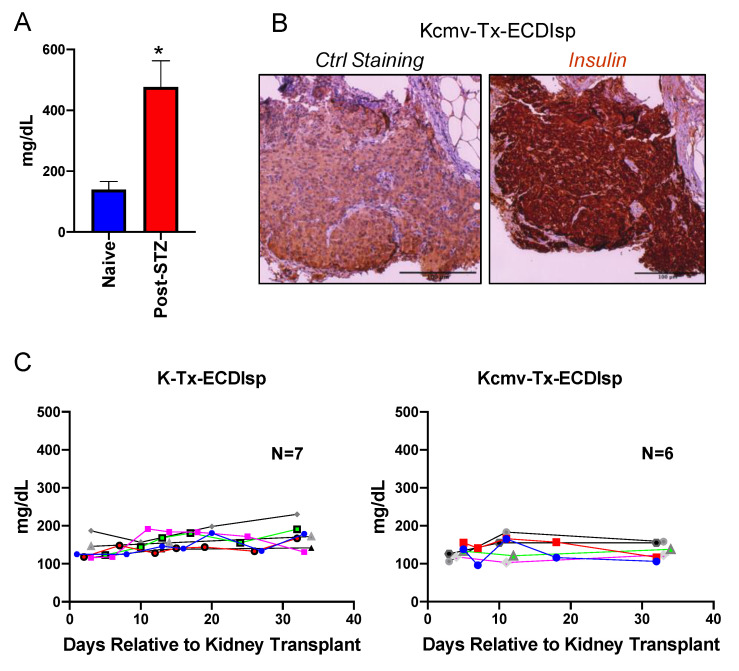
ECDIsp treatment validates continued tolerance of initial islet grafts even after implantation of second latently-infected donor solid organ. (**A**) Blood glucose measurements in naïve mice and mice 3 days post-STZ treatment. (**B**) Representative Insulin staining of transplanted islet grafts 30–35 days post kidney transplantation. Ctrl = Control. Original magnification: 200×. (**C**) Blood glucose measurements demonstrating normoglycemia and islet graft tolerance of mice following kidney transplantation for recipients in ECDIsp+(I)(K-) and ECDIsp+(I)(K+) groups. For (**A**), n = 3 mice per group. For (**B**), Scale Bar is 100 micrometers. * *p* < 0.05.

**Figure 3 pathogens-09-00607-f003:**
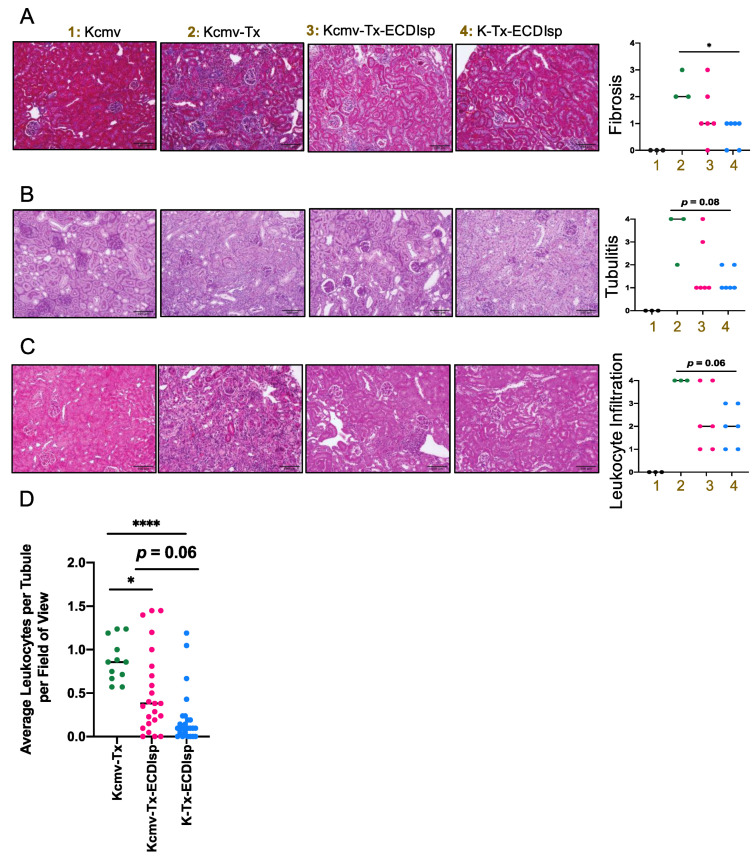
Histologic inflammation in implanted kidneys post ECDIsp, islet transplant, and in latently-infected kidneys. (**A**) Representative Masson’s Trichrome-stained sections of the indicated groups. Bar graph represents a quantitative score of interstitial fibrosis from 0–4, where 4 is most severe. (**B**) Representative Periodic acid-Schiff (PAS) stained sections of polysaccharides proximal to renal tubules to differentiate renal epithelial cells from cell infiltration, as an indication of tubulitis. Bar graph represents a quantitative score of tubulitis. Severity of tubulitis was score by number of infiltrating monocytes in at least 2 tubules per field of view: 0 (no monocytes), 1 (<2 monocytes), 2 (2–4 monocytes), 3 (>4 monocytes, focal), and 4 (>4 monocytes, diffuse). (**C**) Representative H&E stained sections of the indicated groups. Bar graph represents a quantitative score of interstitial inflammation from 0–4, where 4 is most severe. (**D**) Quantification of average number of infiltrating monocytes per renal tubule per 200× field of view. 4 fields of view were examined per mouse PAS histological section with at least 15 renal tubules examined per section. For (**A**–**C**), Original magnification: 200×. Scale Bar is 100 micrometers. N=3–7 mice per group. Statistical significance was determined by Kruskal–Wallis one-way ANOVA. * *p* < 0.05, **** *p* < 0.0001. N = 12–24 fields of view.

**Figure 4 pathogens-09-00607-f004:**
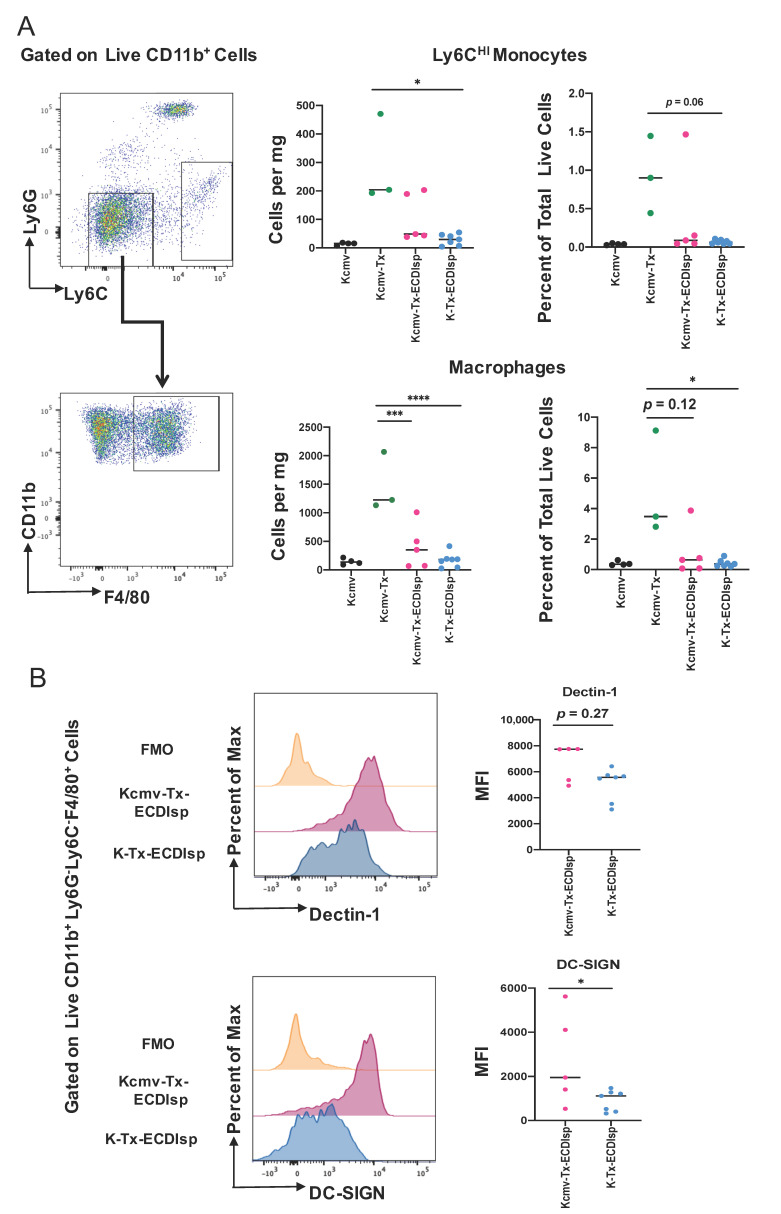
The innate myeloid response in pre-tolerized recipients with latently-infected kidney grafts. (**A**) Representative gating and dot-plots showing percentage of graft or latent donor Ly6C^HI^ monocytes and F4/80^+^ macrophages for each group. Bar graphs quantify total number and frequency of total live cells of Ly6C^HI^ monocytes and F4/80^+^ macrophages. (**B**) Surface expression of Dectin-1 and DC-SIGN on kidney graft macrophages quantified by mean fluorescence intensity at ~30 days post-transplant. MFI levels were analyzed in graft CD11b^+^Ly6G^-^Ly6C^lo^F4/80^+^ cells from Kcmv-Tx-ECDIsp and K-Tx-ECDIsp. Statistical significance was determined by Kruskal–Wallis one-way ANOVA, one-way ANOVA or Mann–Whitney test. * *p* < 0.05, ** *p* < 0.01, *** *p* < 0.001, **** *p* < 0.0001. N = 3–7 mice per group.

**Figure 5 pathogens-09-00607-f005:**
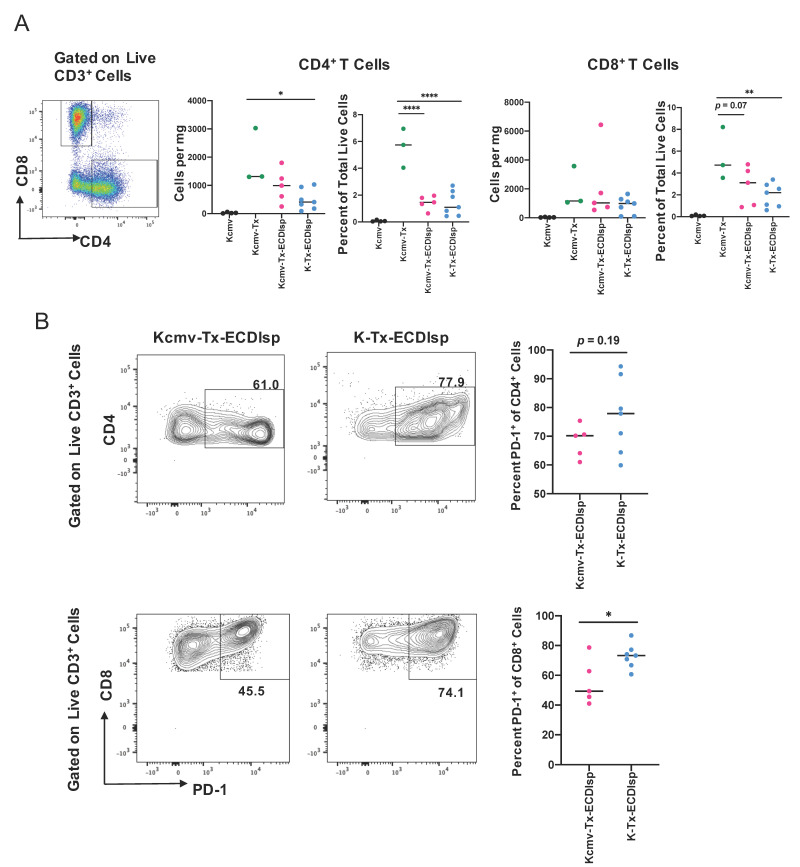
ECDIsp treatment decreases overall T cell accumulation and latently-infected kidneys exhibit reduced PD-1+ CD8^+^ T cells. (**A**) Representative gating and dot-plots showing percentage of graft or latent donor CD4^+^ and CD8^+^ T cells for each group. Bar graphs quantify total number and frequency of total live cells of CD4^+^ and CD8^+^ T cells. (**B**) Percent of PD-1 expressing CD4^+^ and CD8^+^ cells in Kcmv-Tx-ECDIsp and K-Tx-ECDIsp recipients. Representative contour plots show gating for PD-1^+^CD4^+^ cells and PD-1^+^CD8^+^ cells from recipients of each treatment group at ~30 days post kidney transplant. Statistical significance was determined by Kruskal–Wallis one-way ANOVA, one-way ANOVA or student’s *t*-test. * *p* < 0.05, ** *p* < 0.01, *** *p* < 0.001, **** *p* < 0.0001. N = 3–7 mice per group.

**Figure 6 pathogens-09-00607-f006:**
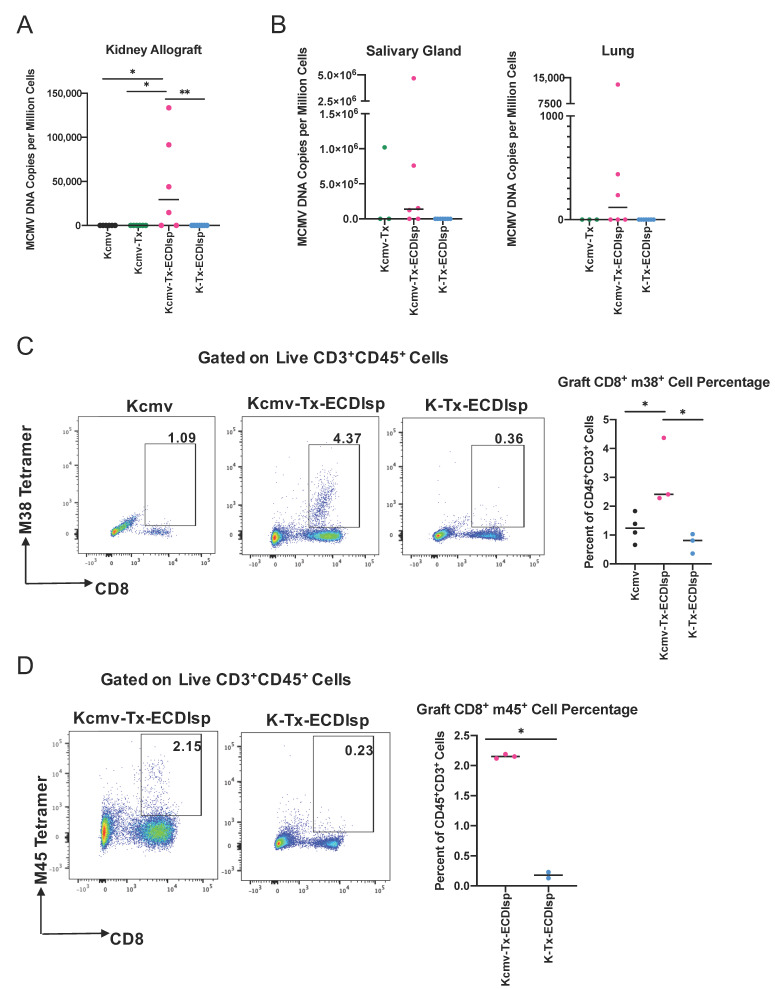
Elevation of viral activation and dissemination markers in pre-tolerized recipients transplanted with latently-infected kidney allografts. (**A**) Quantitative PCR to determine Viral DNA (*IE1)* copy numbers in the grafts of donor contralateral control, Kcmv-Tx-ECDIsp recipients, and K-Tx-ECDIsp recipients. (**B**) Viral DNA (*IE1)* copy numbers in the salivary glands and lungs of Kcmv-Tx-ECDIsp recipients and K-Tx-ECDIsp recipients. (**C**) Representative dot-plot showing percentage of m38 specific CD8^+^ T Cells at ~30 days post kidney transplant. Bar graph denotes the frequency of m38 specific CD8^+^ T Cells in Kcmv-Tx-ECDIsp and K-Tx-ECDIsp grafts. (**D**) Representative dot-plot showing percentage of m45 specific CD8^+^ T Cells. Bar graph denotes the frequency of m45 specific CD8^+^ T Cells in Kcmv-Tx-ECDIsp and K-Tx-ECDIsp grafts. Statistical significance was determined by a Kruskal–Wallis one-way ANOVA, one-way ANOVA or student’s *t*-test. * *p* < 0.05, ** *p* < 0.01. For (**A**,**B**), n = 6–7 mice per group. For (**C**,**D**), n = 2–4 mice per group.

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
