# Peer review of "MCMV Dissemination from Latently-Infected Allografts Following Transplantation into Pre-Tolerized Recipients"

_pathogens, 2020, doi:10.3390/pathogens9080607_

Round 1
Reviewer 1 Report
Manuscript Number: Pathogens-857329
Title: MCMV Dissemination from Latently-Infected Allografts Following Transplantation into Pre-Tolerized Recipients
Article Type: Regular Article
Comments to the authors:
Summary: The paper evaluates MCMV dissemination after transplantation of latently infected kidneys into pre-tolerance-induced recipients. The paper is excellent and very well written. Comments below:
- Please thoroughly check few typographic and punctuation errors.
P7. L249: DECTIN-1, with
- Figure 1. Please use a similar font in A and B. What do the different font colors mean? Please also consider rewriting the legend- for example STZ treatment has not been described.
- Figure 2. What do the different colors in the plots mean?
Reviewer 2 Report
The manuscript submitted by Shah and colleagues presented a highly sophisticated transplantation model to investigate cytomegalovirus reactivation after kidney transplantation. Further, the authors used their model to asked the important question, how a pre-tolerization in an allogeneic transplantation setting affected the reactivation of cytomegalovirus from organ grafts. This question is of high clinical relevance due to the rising number of allogeneic transplantation. Whereas the topic and the model are highly interesting for transplantation physicians the manuscript has some issues that should be solved.
Major points.
- The major problem of the whole manuscript is a very limited number of analyzed animals and time points. I am completely aware that these experiments are technically challenging, time-consuming, and costly. But that is why careful planning and realization is essential and both should be explained in detail to the audience. This is what I really miss throughout the manuscript. The authors described the performed experiments and the results but the rationale for the choose time-frame and technique for analysis is missing.
In detail, I cannot find any explanation throughout the manuscript, why the authors analyzed the virology and immunology of the transplantation at day 30 post-transplantation. As they proposed that the CMV reactivation is mainly triggered early by graft stress during transplantation, 30 days might be much too long for an investigation of virus reactivation and dissemination. After this long time, a primary reactivation has been already controlled by the immune system as demonstrated in the manuscript by CMV-specific CD8 T cell priming. Therefore, it is an important question that has to be discussed, when and to what amount the reactivation takes place, Here the authors are very vague and speculative. - I also do not agree with the statistical evaluation in the manuscript. The authors use only parametric methods that assume a normal distribution. This cannot be assumed with the small sample size. Therefore, it is necessary that the authors should at least comment on why they assume a normal distribution of the data or choose a non-parametric test for analysis.
In particular, it is known that the distribution of viral genomic or infectious viruses in the organ is not normal but log-normal (Fig 6A and B). Also, the dispersion of the presented data is usually so large that no comparable SD can be assumed and therefore a standard t-test or ANOVA will give false results.
Another difficult point to understand is the use of the mean value in all figures, which can lead to an optical overestimation of the few high values. Here I would prefer the median or the authors can dispense with the additional bars and show only the individual animals. - The information given to the used MCMV recombinant is incomplete and a reader not in the field cannot assume why the authors have used this recombinant. Please add the correct reference for your specific recombinant and discuss in more detail, why the authors infected BALB/c mice with an m157 deletion mutant. I know why, but the more common reader might not. Δm157 is not a virus strain, it is a recombinant virus (line 89). Virus strains are e.g. MCMV.Smith or mCMV.K181.
- The authors mentioned that latency was confirmed by serology. But with serology you can only confirm a previous infection with CMV, but not the status of latency. For this it is necessary to analyze the absents of infectious virus in the organs. I think that after seven month the mice are latently infected, but it was not proofed by the authors.
In line with, this I wondered why the authors were not able to detect any viral genomes in the kidneys of the KCMV group (Fig 6A). If this is correct, what is the source for reactivating MCMV.
I also would recommend to introduce in all subfigures of Fig 6 the KCMV and the KCMV+Tx group. Only than it is possible to see if the pre-tolerization has any influence (positive or negative) on CMV reactivation and CD8 T cell priming. - Another important point is the lack of direct evidence for virus reactivation and/or dissemination. The authors did not show reactivation by virus isolation from the organs or a time curse of genome replication in the allograft or the distant target organ. Otherwise someone might speculate that that the source of the detected viral genomes is not the allograft. Therefore, I the authors should more critical discuss what their data show.
- Some parts of the manuscript are hard to understand, due to the number of groups compared in the text. Here another round of proof-reading might help
Minor points:
- In Figure 1B the treatment of the KCMV group is wrong. Following the text, this group should be CMV+.
- In Figure 2 A & B I miss the data of the KCMV+TX group as a positive control for graft rejection. I also would change Fig 2C to 2A as this control was done before 2A and 2B.
- Balb/c is not correct, it is BALB/c
- The histological images are of poor quality in my version of the manuscript and should be enhanced during revision by enlarging or by adding of higher magnifications.
- understand, why it is only important after transplantation. How is the virus purified? Please also comment, what repaired m157 means in Fig. 1A.
- I was wondering, how the authors can correctly assume the severity of monocytic infiltration in Fig. 2, if they count as low as 2 tubules and differentiated between 2, 1, and 0 monocytes in only one cut. Please comment on this.
Round 2
Reviewer 2 Report
The authors have addressed my concerns and definitely improved the manuscript during revision.
I only have one comment left. The figures are hard to read, do to the use of small fonts and points, e.g. 2A-C, 3C, 4, and 5. If possible, please correct this during proof corrections.